# Use of IHF-QD Microscopic Analysis for the Detection of Food Allergenic Components: Peanuts and Wheat Protein

**DOI:** 10.3390/foods9020239

**Published:** 2020-02-23

**Authors:** Ludmila Kalčáková, Bohuslava Tremlová, Matej Pospiech, Martin Hostovský, Dani Dordević, Zdeňka Javůrková, Hana Běhalová, Marie Bartlová

**Affiliations:** 1Department of Plant Origin Foodstuffs Hygiene and Technology, Faculty of Veterinary Hygiene and Ecology, University of Veterinary and Pharmaceutical Sciences Brno, Palackého 1946/1, 612 42 Brno, Czech Republic; lida.anezka.l@gmail.com (L.K.); pospiechm@vfu.cz (M.P.); dordevicd@vfu.cz (D.D.); javurkovaz@vfu.cz (Z.J.); hana.behalova@gmail.com (H.B.); bartlovam@vfu.cz (M.B.); 2Department of Biology and Wildlife Diseases, Faculty of Veterinary Hygiene and Ecology, University of Veterinary and Pharmaceutical Sciences Brno, Palackého 1946/1, 612 42 Brno, Czech Republic; hostovskym@vfu.cz; 3Department of Technology and Organization of Public Catering, South Ural State University, Lenin Prospect 76, 454080 Chelyabinsk, Russia

**Keywords:** quantum dots, QD conjugates, fluorophore, allergen, gliadin, Ara h1

## Abstract

The aim of the study was to analytically evaluate quantum dots in immunohistofluorescence (IHF-QD) microscopic imaging as detectors of food allergens—peanut and wheat. The experiment was designed as two in silico experiments or simulations: (a) models of pastry samples were prepared with the addition of allergenic components (peanut and wheat protein components) and without the addition of allergenic components, and (b) positive and negative commercial samples underwent food allergen detection. The samples from both simulations were tested by the ELISA and IHF-QD microscopic methods. The primary antibodies (secondary antibodies to a rabbit Fc fragment with labeled CdSe/ZnS QD) were labelled at 525, 585, and 655 nm emissions. The use of quantum dots (QDs) has expanded to many science areas and they are also finding use in food allergen detection, as shown in the study. The study indicated that differences between the ELISA and IHF-QD microscopic methods were not observable among experimentally produced pastry samples with and without allergenic components, although differences were observed among commercial samples. The important value of the study is certainly the differences found in the application of different QD conjugates (525, 585, and 655). The highest contrast was found in the application of 585 QD conjugates that can serve for the possible quantification of present food allergens—peanuts and wheat. The study clearly emphasized that QD can be used for the qualitative detection of food allergens and can represent a reliable analytical method for food allergen detection in different food matrixes.

## 1. Introduction

Food allergies significantly affect the quality of life of allergic individuals. An allergic individual requires increased demands on food sources [1]. The labeling of allergens is a crucial tool for obtaining information on the presence of allergenic food ingredients that are intentionally used in a food product. To reduce the risk of adverse food allergic reactions, it is necessary to exclude certain food allergens from the diet. An allergic individual relies on an elimination diet, which is effective only if the food manufacturer informs consumers about the allergen [2,3]. There is growing public interest concerning food allergen labeling due to the increased prevalence of allergies among the worldwide population [1,4,5]. The importance of food allergy as a public health problem has led to its recognition on a global scale of food safety. Guidelines on how to deal with food allergens are provided in Codex Alimentarius and are also embedded in risk management principles and critical control point management systems, which thus offer a framework for prevention in the food industry [2,3,6,7]. Consumers have to be properly informed by food labeling and should not be misled by the provided information, according to Regulation (EU) No. 1169/2011 of the European Parliament and of the Council [8]. 

It is therefore in the public interest to ensure the provision of information to consumers in accordance with food labeling legislation. Reliable methods of detection and quantification of food allergens are necessary to properly monitor compliance [1,9]. The effort to achieve the most reliable identification of food allergens leads to the expansion of detection methods or various modifications made to existing methods. Sensitivity and accuracy of detection methods vary considerably. Therefore, the development of new methods for detecting these substances that achieve more accurate results is necessary [10]. The proposed different approaches to the detection of allergenic ingredients in food products depend on various factors, namely detected allergen, sample matrix, and technological treatments used in food commodity production. For the detection of food allergens, analytical methods based on immunoassays are preferred. Most commercially available kits that are used for routine analysis of food allergens belong to the group of immunological methods. Currently, ELISA is one of the most widely considered and preferred immunoassay methods for detecting food allergens used by the food industry to detect and quantify hidden allergens. The reason these methods are the most common choice for the detection of food allergens is that they are sensitive and specific to the detection of allergenic proteins; moreover, the use of the ELISA method entails a number of advantages such as relatively low cost, rapid application, ease of use, reliability, and speed [11,12,13]. Furthermore, PCR-based methods have also been used successfully for the detection of food allergens, which have been applied for the analysis of heat-treated or otherwise technologically processed foods [14,15,16]. Recently, the number of applications of chromatography methods in conjunction with mass spectrometry (UPLC-MS/MS, HPLC-MS, LC-MS, HR-MS, UHPLC-MS/MS, and others) has grown and they are also widely studied methods in the analysis of food allergens. The methods developed by MS analysis have a lower limit of detection (LOD) due to constant optimization. MS procedures are used to identify and accurately quantify allergenic proteins and peptides [17,18,19,20].

Immunochemical methods, like many other methods, have been further developed, resulting in their various modifications such as immunohistochemical (IHC) or immunofluorescence/immunohistofluorescence (IF/IHF) methods. Chromogens or fluorescent dyes (fluorochromes) are used for allergen labeling in detection by IHF methods. Conventional fluorochromes are relatively sensitive to the external environment, including normal imaging conditions [21]. In terms of the fluorescence life of conventional fluorochromes, organic semiconductor nanoparticles—quantum dots (QDs)—appear to be suitable for immunoanalytical labeling [22,23]. Quantum dots are inorganic semiconductor fluorescent nanoparticles with a size in the range of several nm, the most common arrangement being the so-called core–shell. The core consists of one type of semiconductor (e.g., CdSe; CdTe), and the shell consists of several layers of atoms of the second type of semiconductor (e.g., ZnS; CdS), forming a protective structure around the shell [24]. Due to the outer coating, solvation in aqueous solution is possible. The sheath also acts as a carrier of the reactive group R, which is necessary for bioconjugation [24]. This is very important because they are used as a fluorescent tool for labeling of countless biologically active substances, especially in the form of so-called bioconjugates [25]. Quantum dots possess exceptional photophysical properties, such as dimensionally tuned narrow and symmetric emission, broad and strong quantum yield, and high stability to photochemical and chemical radiation, and thus occupy a new class in the group of inorganic fluorescent labeling tools [23,25,26,27]. In addition to their optical properties, CdSe/ZnS QDs, which consist of a CdSe core and a ZnS sheath that increase the QDs’ resistance to light and increase their quantum yield, are the most popular among scientists [28,29]. The use of such QDs is widespread in the life sciences, but several publications have reported the use of QDs in the food sciences, where they were mostly used to detect pathogenic bacteria, proteins, and rotaviruses. They were also used in applying polyclonal antibody-bound QDs and to intensify the signal of detecting allergens [23,25,26,27,29,30,31,32].

The aim of this work was to develop a suitable IHF-QD methodology using quantum dots as a new type of fluorophore in immunohistofluorescence labeling of allergens in different food matrixes. Specifically, we sought to find out the possibilities of using quantum dots as a labeling tool suitable for the IHF-QD microscopic analysis of allergens, namely peanut and wheat protein components.

## 2. Materials and Methods

### 2.1. Analyzed Material

For the purposes of Simulation 1, model pastry samples with the addition of an allergenic component were used as positive model samples (8 with the addition of a peanut ingredient, AraK1+ to AraK8+; 8 with the addition of a wheat protein component, GK1+ to GK8+). For the negative model samples, no allergenic ingredients were used (AraK1- and AraK2-; GK1- and GK2-), as we can see in Table 1. Individual K + controls were prepared by adding an allergenic component in ascending percentage from 0.01% to 10%. Specifically, ground peanuts (ALIKA a.s., Čelčice, Czech Republic) were used to analyze the peanut allergenic component. Wheat protein (Amylon a.s., Havlíčkův Brod, Czech Republic) was added to the positive model samples for analysis of the wheat allergenic component. Model samples were prepared with the addition of wheat protein or ground peanut seeds in increased concentration from 0.01% to 10%. The wheat protein or ground peanut seed in desired concentration was added to dough (consisting of sugar, corn flour, egg, milk, and baking powder). Pastry dough was prepared as follows: for 0. 01% concentration, 99.99 g of dough was used and 0.01 g of wheat protein or ground peanut was added; for the concentration of 0.1%, 99.9 g of dough and 0.1 g of wheat protein or ground peanut were used; for the concentration of 1%, 99 g of dough and 1 g of wheat protein or ground peanut were used; and for the concentration of 10%, 90 g of dough and 10 g of wheat protein or ground peanut were used. The dough was stirred using a dough kneader. Dough with one rate of concentration was split into smaller parts of fist size and baked at 180 °C for 15 min in an oven. For the purposes of Simulation 2, commercial food products were used. As a commercial food product, the following selected groups of products for special nutrition—pastries, breakfast cereals, and protein, cereal, and raw bars—were chosen for immunohistofluorescence examinations. Negative commercial samples were considered to be those that had no declared allergenic ingredients in the list of ingredients. Positive samples were those that had declared allergenic components.

### 2.2. Preparation and Processing of Samples

Four representative 1 cm^3^ samples were taken from each model or commercial sample. A formaldehyde, acetic acid, ethanol, and distilled water (AFA) fixing solution was used to maintain the structures [33]. Histological preparations of model and commercial food samples were processed using a paraffin block technique. This technique involves individual steps such as dewatering and saturation of samples, paraffin embedding, subsequent cutting of the blocks into histological sections, and dewaxing of sections prior to the IHF-QD microscopic method. First, it was important to remove the AFA fixative from the samples by washing with distilled water. This was followed by dewatering of the samples in an AT-4 (Leica, Wetzlar, Germany) autotechnicon by ascending alcohol (ethanol) series and xylene. The dewatered samples were further saturated with warmed (56–58 °C) paraffin in the autotechnicon. Then, samples were embedded on embedding liquid paraffin to obtain paraffin blocks. Four paraffin blocks (A, B, C, D) were generated for each sample. The obtained paraffin blocks were cut on a rotary microtome RM 2255 (Leica, Wetzlar, Germany) to a thickness of 5 µm. The slicing into smooth sections was followed by the tensioning of the sections’ the water level, which is part of the microtome. Individual sections were captured on SuperFrost Plus slides (Thermo Fisher Scientific, Waltham, MA, USA). Next, sections were dried on a plate heated to 40 °C. The samples were then placed in a thermostat at a constant temperature of 50 °C for 24 h. Dewaxing was performed in xylene and alcohol baths. For the antigen-retrieval step, citrate EDTA (wheat samples) or AlCl3 (peanut samples) were used. Our methodology also included permeabilization in 0.25% Triton X-100, and to prevent nonspecific binding in our method we used blocking buffer 6% BSA with goat normal serum. The preparation of samples was the same for Simulations 1 and 2.

### 2.3. IHF-QD Microscopic Method

A modified immunofluorescence method was used for the detection of selected allergens in food samples. The modified method is based on the principles outlined in the quantum dot manufacturer’s guide (Invitrogen, Carlsbad, CA, USA). The basis of this method is an indirect two-stage immunohistochemical method. For antigen retrieval, a buffer, namely 4% AlCl3 buffer adjusted to pH 3.5 (peanut) and citrate EDTA adjusted to pH 6.2 (wheat protein), was applied according to the type of allergenic component. The samples in buffer solution were microwave heated at 650 W for 5 min, then cooled at room temperature for 20 min. The next step consisted of blocking the endogenous peroxidase activity with a 4% formaldehyde solution in PBS. A solution consisted of 3 types of another solutions (PBS + BSA with sodium azid (Thermo Fisher Scientific, Waltham, MA, USA) + goat normal serum) were used to block non-specific binding. Specific polyclonal primary antibodies of rabbit origin anti-ara h1 (INDOOR biotechnologies, Charlottesville, WV, USA) and anti-gliadin (Sigma-Aldrich Company, St. Louis, MI, USA) diluted 1:500 with antibody diluent (DakoCytomation ref. S0809) were used. Incubation of contained antigen with primary antibody was run for 60 min. Commercially purchased secondary antibodies to a rabbit Fc fragment with labeled CdSe/ZnS QD at 525, 585, and 655 nm emissions were used to label primary antibodies. After the IHF-QD method, the slides were equipped with the commonly used SOLAKRYL mounting medium. Sample sectional analyzes were performed with a Leica DM 3000 fluorescence microscope (Leica, Wetzlar, Germany) using fluorescent filters recommended by the quantum dot manufacturer (Chroma Technology Corp., Bellows Falls, VT, USA). The field of view was captured by a Leica DFC 295 camera (Leica, Wetzlar, Germany). The visual field acquisition was performed using a Leica DFC 295 camera (Leica, Wetzlar, Germany) in conjunction with Xn View computer software (1.97.8, Pierre E. Gougelet, Reims, France), and images were further analyzed using NIS Elements BR version 4.50.00 (Nikon Instruments Inc., Melville, NY, USA).

For Simulation 1, that is, the development of the new IHF-QD method, 10 model samples were used for each allergen component, namely 8 guaranteed positive samples and 2 guaranteed negative samples. Four blocks were prepared from each sample, with 8 slices per IHF-QD analysis per block. Each analysis was carried out anonymously in a total of three consecutive analyzes. IHF-QD was performed identically for all three types of conjugates, namely QD 525, 585, and 655 (hereafter referred to as IHF-QD 525, IHF-QD 585, and IHF-QD 655). Samples were marked as positive (≥85% confidence), doubtful (around 50%) and negative (≤5%) (Table 2). Spectral analysis (Simulation 1) was evaluated using NIS Elements BR with mean intensity setting. Eight guaranteed positive samples were analyzed, 10 images were selected for each sample, and the mean intensity for each type of QD was further evaluated. A K0 control was created for each sample to eliminate false positives as well as to compare background intensity. Each spectral image consisted of many channels, each representing one wavelength. A histogram was created for each image and the mean intensity value was derived from the intensity histogram. Mean intensity is the arithmetic mean of pixel intensities.

Simulation 2 consisted of the comparison of the ELISA detection method with our IHF-QD microscopy detection method (QD with an emission wavelength of 585 nm) and the manufacturer’s declaration of the contained ingredients. Each analysis tested 20 samples for the peanut allergenic component and 20 samples for the wheat allergenic component. Sample detection was carried out anonymously. 

### 2.4. ELISA Method

Two types of commercial kits were used—RIDASCREEN Gliadin and RIDASCREEN FAST Peanut (R-Biopharm AG, Darmstadt, Germany)—and were always applied according to the planned detected target group of allergenic ingredients.

ELISA kit RIDASCREEN Gliadin (Art. No.: R7001) is an immunoassay with a sandwich enzyme for the quantitative analysis of prolamins from wheat (gliadin), rye (secalin), and barley (hordein). It should be used to control the contamination of dietary products in celiac patients. It is approved by the AOAC as the official method of the first action. The method has been assigned the official AOAC method number 2012.01. The test kit is also a method tested for performance by the AOAC Research Institute (AOAC-RI 120601).

RIDASCREEN FAST Peanut (Art. No: R6202) is a sandwich enzyme immunoassay developed for the quantitative analysis of peanuts and peanuts in food. As an example for the pastry and candy food groups, the following commodities were tested: breakfast cereals, cookies, ice cream, and milk chocolate. The ELISA has been approved by the AOAC Performance Tested Method Program and awarded the PTM Certificate No. 030404 by the AOAC Research Institute. The test may also be used to analyze other food samples.

Quantification of allergenic components depends on the comparison of tested antigen responses with reactions of a number of standard dilutions. Thus, a typical standard curve for RIDASCREEN is used for quantification. The quantitative results of the ELISA methods were obtained by measuring the ELISA reader according to the instructions of the ELISA kit manufacturer and the associated software that can automatically evaluate the content of a given allergen in mg/kg based on standard measurements. The gluten calculation is based on the assumption of a 1:1 ratio between gliadin and glutenin, wherein the total gluten can be expressed in mg/kg. The results for the peanut allergen are expressed in mg/kg of peanut. The samples were anonymously coded.

### 2.5. Statistical Analysis

The obtained results were further processed using mathematical and statistical methods using MS Excel 2016 (Microsoft Corporation, Redmond, WA, USA) and Unistat 6.1 (Unistat Ltd., London, UK). In Simulation 1, the results of three types of quantum dot conjugates (with emissions of 525, 585, and 655 nm) were analyzed as a statistical consensus/conformity with the ELISA method. To evaluate the commercial sample testing using the IHF-QD and ELISA methods (Simulation 2), McNemar’s test [34] utilizing the test criterion calculation of χ2 (chi quadrate) was used. The absolute numbers of positive and negative samples were used for McNemar’s test. In the case of a doubtful result (IHF-QD microscopic method), analysis was performed as if there were two samples—one with a negative result and the other with a positive result. Statistical significance at *p* < 0.05 was evaluated by one-way ANOVA analysis of variance, and parametric Tukey’s post hoc test (when Levene’s test showed equal variances *p* > 0.05) and nonparametric Games–Howel post hoc test (when Levene’s test showed unequal variances *p* < 0.05). Statistical software SPSS 20 (IBM Corporation, Armonk, NY, USA) was used.

## 3. Results and Discussion

### 3.1. Application of IHF-QD Method to Model Samples and Comparison of Results with ELISA Method (Simulation 1)

**Simulation 1 focused on the possibility of evaluating samples using three types of quantum dots.** IHF-QD microscopic analysis gives us qualitative results about the presence or absence of an allergenic component. Qualitative evaluation was evaluated as positive, negative, or doubtful. To verify the plausibility of the results obtained by the IHF-QD method, model samples were also analyzed by the sandwich ELISA method (Table 2).

Individual qualitative results of IHF-QD methods were compared with those of the ELISA method, whose results corresponded to the content of analyzed model samples. Model samples (AraK1-, AraK2-, GK1-, GK2-) free of allergenic components were used to demonstrate that the conjugates used bind only to specific allergenic fractions of the food matrix. As we can see in Table 2, in our experiment no false positive detection was detected in any of the tested methods. Conformity testing showed that the ELISA results did not differ from the IHF-QD 585 results, that is, the IHF-QD 585 achieved 100% compliance with the ELISA method. While the testing of the IHF-QD 585 method was consistent with the comparison method, it should be noted that the remaining methods, IHF-QD 525 and IHF-QD 655, did not achieve this level of compliance. For IHF-QD 525 and IHF-QD 655, doubtful results were evaluated in four and two samples. Doubtful results were evaluated based on the contrast intensity between the labeled allergen component QDs and the background that was not high enough. Because of this observed phenomenon, the contrast between conjugates with different types of QDs was compared.

The mean intensity of the IHF-QD microscopic detection of the peanut and wheat protein components was evaluated. The resulting mean intensities (Figure 1 and Figure 2) pointed to the fact that conjugates of 585 QD showed the highest contrast intensities against the background. The statistically significant (*p* < 0.05) difference in mean intensities, especially between samples with allergens (peanuts and wheat protein) and without them (control samples), can be observed in Figure 1 and Figure 2. However, high standard deviations emphasize the complexity of food allergen detection (Figure 1 and Figure 2). Statistically significant (*p* < 0.05) differences, both for peanut and wheat protein, were checked by one-way ANOVA (nominal results) and obtained results show no differences (*p* < 0.05) between ELISA results and IHF-QD results. These results support the hypothesis about the IHF-QD method’s reliability.

Figure 3 and Figure 4 illustrate the actual view perceived by the evaluator’s eye. The results show that the IHF-QD microscopic method using QDs with 585 nm emission achieves the highest contrast intensity compared to the same method using QDs with emissions of 525 nm or 655 nm. The intensity of the light signal may vary between types of QDs, as confirmed by Byers and Hitchman [35], who state that the light signal with respect to its intensity varies between types of QDs, for example, the green QD signal (525 nm) is 17 times lower than the intensity of the red QD signal (655 nm). However, this also depends on the type of method processing, as well as the (microscopic) detection technique, that is, fluorescence microscope vs. a confocal laser scanning microscope or others. There is a link between the structure, size of applied quantum dots versus pH, and the chemical composition of buffers and other reagents [36,37,38]. Our results indicated that QDs with 585 nm emissions are the most suitable for the IHF-QD method we tested.

The possibility of applying quantum dots attached to a polyclonal antibody amplifying the light signal from allergen detection instead of the routine use of fluorochromes [31] has been investigated in recent years. In our research, we evaluated the allergen detection suitability using QDs bound to secondary antibodies in immunohistofluorescence microscopic methods. QDs appear to be an excellent tool for labeling allergenic ingredients in IHF-QD microscopic methods. This claim is also supported by the fact that, compared to conventional fluorescent dyes, QDs are extremely stable and can be subjected to repeated cycles of excitation and fluorescence for several hours to days while maintaining a high level of brightness. For QDs, light with a half-life of several tens of nanoseconds (30–100 ns) is emitted at room temperature. Emission is slower than autofluorescence background degradation. Moreover, the advantage of QDs is their resistance to photobleaching over most fluorescent agents in which photostability is a critical parameter [26]. Due to the photostability, quantum dots have a great potential for use in fluorescence microscopy. The use of QDs has been also verified in research on developing imaging methods of mouse fibroblasts, where QDs gave a high signal-to-background ratio [26], the same as in immunoanalytical analyses [23,24,25]. QD fluorescent labeling has been widely used to detect a number of food grade substances such as vanillin [39], allergens such as alpha lactalbumin [21], mycotoxins [40], or pathogens [41,42].

With respect to fluorescent imaging of food components, such as gluten and peanuts, cases were confirmed by a number of scientific studies. For example, one study examined the visualization of the gluten network in flat bread and zein in corn extrudates by confocal laser scanning microscopy [43], using a CdSe/ZnS QD labeling tool with an emission of 620 nm, using the method of covalent bioconjugation principles with immunohistochemistry rules.

Additionally, further CdSe/ZnS quantum dots were used in a study by a team of scientists, Ansari et al. [44]. They used quantum dots conjugated to gliadin antibodies to monitor the molecular distribution of gliadin proteins in raw dough samples and the molecular distribution during the bread baking. In that experiment, the specificity of gliadin antibodies was demonstrated by Western blot using confocal laser scanning microscopy as an imaging technique. Bonilla et al. [45] developed an immunoassay method with three types of quantum dots (with emissions of 525, 585, and 655 nm). They used QDs in the microscopic detection of food components, specifically to visualize the distribution of low-molecular-weight glutenins, high-molecular-weight glutenins, and gliadins simultaneously. Each type of dot was used to immuno-label a different fraction of gluten protein. Detection was performed using a confocal scanning microscope. To detect peanuts as an allergenic component in food, quantum dots were used in an immunoassay using a biosensor strategy using Qdots-aptamer-GO complexes as probes. Using this biosensor with QDs, the main peanut allergen Ara h1 with high sensitivity and selectivity on a miniaturized optical detector was detected. A biscuit sample was also analyzed in which Ara h 1 was detected by the assay and commercial ELISA kit for comparison, and the results showed that the detection method was very fast, sensitive, and reliable [46].

### 3.2. Comparison of ELISA Detection, IHF-QD Microscopy Detection, and Manufacturer’s Declaration (Simulation 2)

The IHF-QD microscope method using QD 585 was further used in the study to detect peanut and wheat protein allergenic components in market samples. The distribution according to the allergenic component content can be seen in Table 3. ELISA methods were used as the reference methods. The methods used were carried out using commercial quantitative assay kits. ELISA methods have been validated by the AOAC (Art. No.: R7001; R6202) and are intended to quantify allergenic ingredients from food samples. Based on the ELISA results, it was possible to compare the detected results of the detection of allergenic components using the IHF-QD 585 microscopic method, as well as the truthfulness of the declaration of the content or absence of the allergenic component (Table 3). 

The standard curve was used to calculate the levels of allergenic components in each sample. Using the ELISA method, it was evaluated whether there was zero content, there were trace amounts, or there were more than trace amounts. For gluten, the maximum value considered to be zero content was 20 mg/kg, and the maximum value was considered to be a trace amount of 50 mg/kg. For peanuts, the maximum level determined as zero was considered to be 2.5 mg/kg, and the trace level was considered to be 25 mg/kg [8,47]. As shown in Table 3, 25% of the samples analyzed (10 out of 40) did not comply with the manufacturer’s declaration. In addition, the results of the study by Pele et al. (2007) point out that, after ELISA analysis, almost 25% of the analyzed products, with no declaration about peanut content, showed that they contained peanut [48]. However, the EU Consumer Food Regulation No. 1169/2011 strictly requires that allergen information be provided to customers in different ways—either on the menu, on the notice board, orally communicated to employees, or in other formats. If communicated orally, it must be clearly explained to customers how they can obtain access to the information [49].

The obtained results demonstrate the low quality of food safety in terms of the management of allergenic ingredients in food establishments. An allergic individual is dependent on trust in food businesses [3]. An allergen that is not labeled as well as unintentional allergen contamination of foods are examples of serious threats to human health. Despite a few promising treatment strategies in the area of food allergies, treatment is also largely based on avoiding allergen-containing foods. This entails checking the composition of individual food products at the time of purchase, that is, reading the labels for each purchase, which entails knowledge of labeling laws (which vary by country) and of warnings to avoid products. The patient’s safety depends on their level of education and also on their risk-free behavior. Risk behavior can lead to unexpected reactions [50,51]. Acute allergic reactions to food account for a high proportion of hospital admissions [1]. The allergic reaction can vary from very mild to severe and in some cases even fatal [51], depending on the dose, the individual, and other factors. The problem of food allergy is also associated with a plethora of medical visits, and even cases of death have been reported [52].

The level of producers’ allergen management depends on the knowledge and work discipline of the employees as well as the level of company environment [3]. There have been many accidental cross-contaminations of additives into industrially processed food containing allergenic ingredients at dangerous levels or cases of deliberate adulteration for economic purposes [53]. The numbers of allergic patients with food allergies are steadily increasing. Prevalence affects up to 4% of adults and 6% of children worldwide [49]. According to Bartuzi et al. [54], an incidence of food allergies is 4.3% in individuals 0–17 years old, with the highest incidence in the youngest age group. Furthermore, he reported that food allergy was reported in Poland in 2006–2008 among 13.0% of individuals in the age group 6 to 7 years, 11% in the age group 13 to 14 years, and 5.0% among adults. The American population has allergen prevalence from 3.5% to 4% of the total population, with 4% for adults and 5% to 8% for children. There have been up to 30,000 hospitalizations per year in the USA due to a food allergy [51]. Thyagarajan and Burks [55] estimated that 6% of children under 3 years old and 4% of adults are affected by a food allergy. Of the approximately 125,000 emergency visits in the United States, food allergy is associated with about 15,000 for which secondary food-induced anaphylaxis occurs. Hidden allergens as part of food products pose a very significant problem, and due to the undeclared allergen content, foods are often withdrawn from the market. Accidental and unaware consumption posed a significant health threat to more than 50% of peanut allergic people and 30% of nut allergic children. According to Soon [49], peanuts are among the top 10 foods responsible for most food allergies in the UK. Peanut allergy in children in the UK is around 1.5%. Every year in the UK, 10 patients die from anaphylaxis caused by the consumption of food for which no allergenic ingredients were reported. The prevalence of food IgE sensitization by peanuts among European states ranges from 0.45% (Reykjavik) to 7.18% (Madrid) and by wheat from 0.67% (Reykjavik) to 10.47% (Madrid). The allergies to peanut and wheat proteins are the most common allergies among the worldwide population [56,57,58]. Zhou et al. (2019) report that the prevalence of allergy to wheat is 3.0–4.2% in Europe [17]. Consumers are totally dependent on the availability, accuracy, and quality of the food information of products they want to buy [1,49]. Doctors and nutritional specialists recommend to allergic people that they avoid foods labeled as “may contain” allergens [51], but it should also be noted that the use of the warning “May contain traces of allergen…” only as a precautionary warning has become so widespread that it is very difficult to find examples of certain groups of food products without them. A survey conducted in the United States and Canada stated that allergic consumers mistakenly believe that such labels are regulated. These consumers interpret the risk they perceive when reading these terms, with 11% buying “may contain” and 40% buying “at the facility they also process”, although there is no difference in actual risk between these interpretations [49]. Allen and Taylor [59] stated that these labels are confusing for consumers and reduce the quality of life of allergic individuals. It is known that many allergic consumers ignore these claims of potential content. Furthermore, Allen and Taylor [59] reported that there are analytical surveys suggesting that many of the products labeled “may contain traces” contain no detectable allergen residues and are likely to be safe for allergic consumers. However, on the contrary, there are other authors reporting opposite suggestions.

Table 3 also shows the results of two methods of our experiment applied to products from the market network. The results of Table 3 were used to verify and compare IHF-QD 585 with ELISA. The correlation between the IHF-QD microscopic method and the ELISA method was determined to be 73%. McNemar’s Test, a test on a 2 × 2 contingency table, was used. Two groups—ELISA results and IHF-QD results—were used as paired data. Chi squared equals 0.250 with 1 degree of freedom. The *p*-value was *p* > 0.01.

Statistically significant (*p* < 0.05) differences were not observed between results obtained by ELISA and IHF-QD 585 tests for the market samples.

Quantum dot (QD) conjugates can be used for many immunohistochemical applications. They have good optical, excitation/emission, and photostable properties and offer many advantages over the use of chromogens or organic fluorophores in these applications [60]. 

## 4. Conclusions

The IHF-QD method applied to the analysis of two types of allergens was compared with routinely used ELISA methods. Quantum dots can be considered as a suitable fluorescent tool for labeling allergens in the food matrix. The emission intensity of QDs, which gave a high contrast to the background in the application of 585 QD conjugates, was also evaluated. High contrast can be used mainly for quantification of the result, especially by image analysis for future research studies. This IHF-QD microscopic method can be considered a reliable tool for the rapid detection of gliadin and peanut in commercial food products. PCR and ELISA methods also represent reliable methods for allergen detection. The IHF-QD microscopic method for allergen detection in comparison with PCR and ELISA methods can be evaluated by looking at four major elements: the price, whether it is time consuming, technical skills, and laboratory equipment. The disadvantages of the IHF-QD method are the price and the fact that it is time consuming. On the other hand, the IHF-QD method may be more adequate for laboratories having a fluorescence microscope and the technical skills needed for this method. Another advantage of the IHF-QD method is the possibility of storing already evaluated samples that can be evaluated repeatedly over the next few weeks. This experiment can be followed up by further studies to detect other food allergens, and the application of bioconjugates with QDs seems to be suitable for multi-labeling procedures. The truthfulness of the declaration on the content or absence of allergenic components was also evaluated in this study. In 25% of the analyzed foods, the declaration was not true. It is therefore necessary to impose increased demands on the allergen management for food products. Furthermore, these facts stress the importance of our study and of the information that it provides.

## Figures and Tables

**Figure 1 foods-09-00239-f001:**
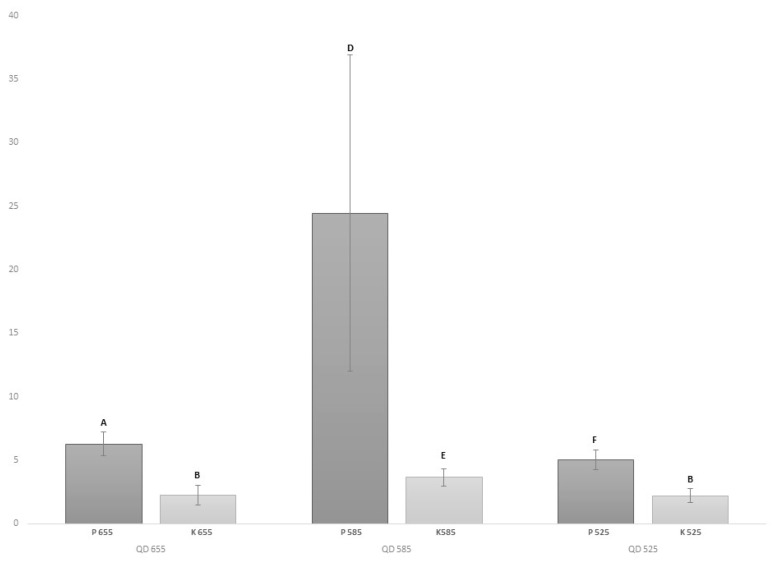
Comparison of contrast between conjugates with different types of QDs. Average rate of total contrast intensity for gliadin. Note: P: positive sample; K: control. Different letters (A, B, C, D, E, F) indicate statistically significant (*p* < 0.05) differences.

**Figure 2 foods-09-00239-f002:**
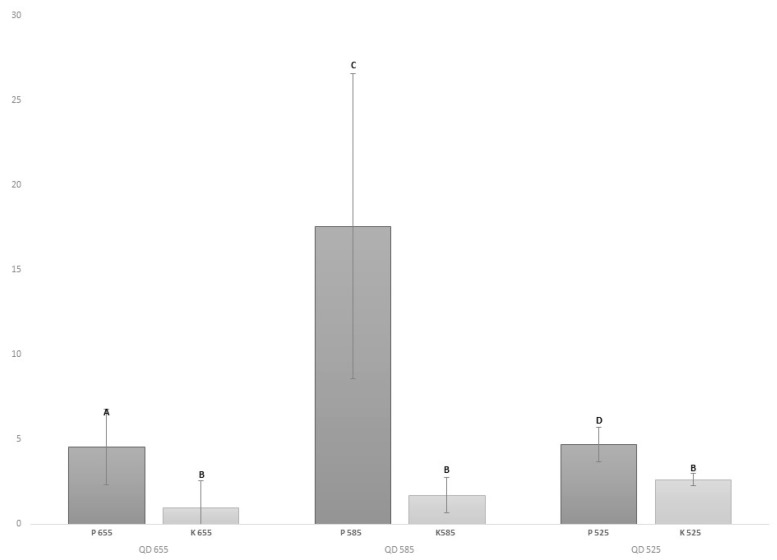
Comparison of contrast between conjugates with different types of QDs. Average rate of total contrast intensity for Ara h1. Note: P: positive sample; K: control. Different letters (A, B, C, D, E, F) indicate statistically significant (*p* < 0.05) differences.

**Figure 3 foods-09-00239-f003:**
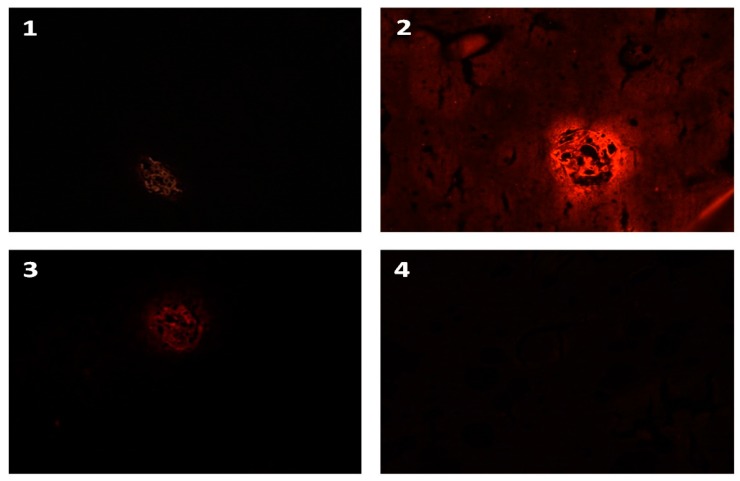
IHF-QD microscopy imaging of the wheat protein component (model sample GK7+). Note: **1**: 525 QD conjugate; **2**: 585 QD conjugate; **3**: 655 QD conjugate; **4**: negative control.

**Figure 4 foods-09-00239-f004:**
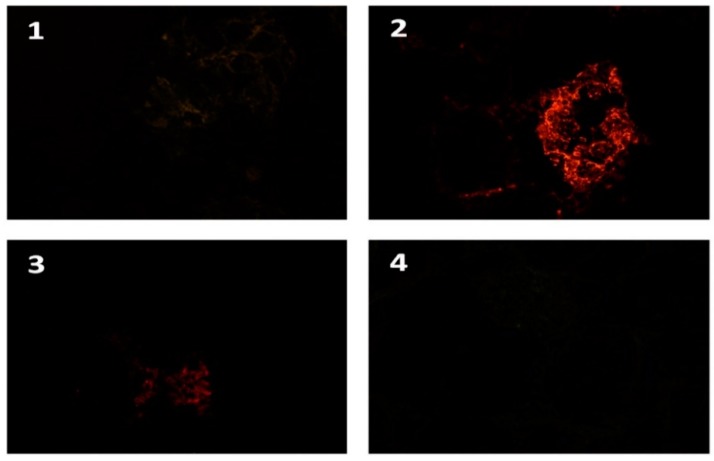
IHF-QD microscopy imaging of the peanut component (AraK7+). Note: **1**: 525 QD conjugate; **2**: 585 QD conjugate; **3**: 655 QD conjugate; **4**: negative control.

**Table 1 foods-09-00239-t001:** List of model samples.

Sample Code	Addition of Allergenic Component (%)	Sample Code	Addition of Allergenic Component (%)
AraK1+	0.01	GK1+	0.01
AraK2+	0.01	GK2+	0.01
AraK3+	0.1	GK3+	0.1
AraK4+	0.1	GK4+	0.1
AraK5+	1.0	GK5+	1.0
AraK6+	1.0	GK6+	1.0
AraK7+	10.0	GK7+	10.0
AraK8+	10.0	GK8+	10.0
AraK1-	0.0	GK1-	0.0
AraK2-	0.0	GK2-	0.0

**Table 2 foods-09-00239-t002:** Comparison of the qualitative results of the ELISA method with those of the quantum dots in immunohistofluorescence (IHF-QD) method using different types of QD (QDs of different wavelengths) model samples.

Sample Code	Allergenic Component Content	ELISA Result	IHF-QD Result		
			525	585	655
AraK1+	✔	✔	✔	✔	✔
AraK2+	✔	✔	✔/**×**	✔	✔/**×**
AraK3+	✔	✔	✔/**×**	✔	✔
AraK4+	✔	✔	✔	✔	✔
AraK5+	✔	✔	✔	✔	✔
AraK6+	✔	✔	✔	✔	✔
AraK7+	✔	✔	✔	✔	✔
AraK8+	✔	✔	✔	✔	✔
AraK1-	**×**	**×**	**×**	**×**	**×**
AraK2-	**×**	**×**	**×**	**×**	**×**
GK1+	✔	✔	✔/**×**	✔	✔
GK2+	✔	✔	✔	✔	✔
GK3+	✔	✔	✔	✔	✔
GK4+	✔	✔	✔	✔	✔/**×**
GK5+	✔	✔	✔/**×**	✔	✔
GK6+	✔	✔	✔	✔	✔
GK7+	✔	✔	✔	✔	✔
GK8+	✔	✔	✔	✔	✔
GK1-	**×**	**×**	**×**	**×**	**×**
GK2-	**×**	**×**	**×**	**×**	**×**

Note: positive ✔, doubtful ✔/**×**, negative **×**

**Table 3 foods-09-00239-t003:** Sample analysis collected from the market network.

Sample Code	Manufacturer’s Declaration	ELISA Result (Gluten Content (mg/kg))		IHF-QD 585 Result
Ara1	without content	✔	trace amount (23.84)	✔
Ara2	without content	**×**	zero content * (0.09)	**×**
Ara3	without content	✔	more than a trace amount ** (>25)	**×**
Ara4	it may contain trace amounts	**×**	zero content (1.21)	**×**
Ara5	it may contain trace amounts	**×**	zero content (2.13)	**×**
Ara6	it may contain trace amounts	✔	more than a trace amount (>25)	✔
Ara7	it may contain trace amounts	✔	trace amount (7.33)	**×**
Ara8	without content	**×**	zero content (1.60)	**×**
Ara9	without content	**×**	zero content (0.07)	**×**
Ara10	it may contain trace amounts	✔	more than a trace amount (>25)	✔
Ara11	content	✔	more than a trace amount (>25)	✔
Ara12	content	✔	more than a trace amount (>25)	✔
Ara13	content	✔	more than a trace amount (>25)	✔
Ara14	content	✔	more than a trace amount (>25)	✔
Ara15	content	✔	more than a trace amount (>25)	✔
Ara16	content	✔	trace amount (16.61)	✔
Ara17	content	✔	more than a trace amount (>25)	✔
Ara18	content	✔	more than a trace amount (>25)	✔
Ara19	content	✔	trace amount (10.94)	✔
Ara20	content	✔	trace amount (21.83)	✔
G1	without content	**×**	zero content (11.39)	✔
G2	without content	**×**	zero content (13.20)	**×**
G3	without content	✔	trace amount (34.79)	✔
G4	without content	✔	more than a trace amount (55.03)	✔
G5	it may contain trace amounts	✔	more than a trace amount (> 80)	✔
G6	without content	✔	more than a trace amount (58.45)	✔
G7	it may contain trace amounts	✔	more than a trace amount (63.68)	✔
G8	without content	✔	more than a trace amount (54.15)	✔
G9	without content	**×**	zero content (17.19)	**×**
G10	without content	**×**	zero content (11.02)	**×**
G11	content	✔	trace amount (42.82)	✔
G12	content	✔	trace amount (36.96)	✔
G13	content	✔	more than a trace amount (68.63)	✔
G14	content	✔	more than a trace amount (73.17)	✔
G15	content	✔	more than a trace amount (63.32)	✔
G16	content	✔	more than a trace amount (57.46)	**×**
G17	content	✔	more than a trace amount (75.10)	✔
G18	content	✔	more than a trace amount (74.70)	✔
G19	content	✔	more than a trace amount (> 80)	✔
G20	content	✔	more than a trace amount	✔

Note: positive ✔, doubtful ✔/×, negative ×. * Note: Under limit of detection (<LOD). LOD Ara: 0.03 –0.13 mg/kg. LOD G: 1.2–1.5 mg/kg gliadin (2,5 mg/kg gliadin corresponding to 5 mg/kg gluten). ** Note: Over the limit of quantification (>LOQ). LOQ Ara: 20–25 mg/kg. LOQ G: 80 mg/kg

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
