# Peer review of "Use of IHF-QD Microscopic Analysis for the Detection of Food Allergenic Components: Peanuts and Wheat Protein"

_foods, 2020, doi:10.3390/foods9020239_

Round 1

Reviewer 1 Report

Overall comments:

In this manuscript, Ludmilia et al. report on the use of quantum dots in immunohistofluorescence (IHF-QD) to detect allergens in food samples. Food allergen detection is a major issue in public health and new approaches are welcome.  While the authors appear successful in this endeavour, they fail to provide quantitative comparisons of their IHF-QD technique over existing methodologies. Quantum dots have been used previously to detect food allergens (including some of the same allergens presented in this study), and without the quantitative comparison, the utility of the paper is questionable as it appears to be repetetive.

While the authors accurately identify several advantages of quantum dots over traditional fluorophores (increased quantum yield, stability etc.), they fail to provide any quantitative comparison to demonstrate these advantages when applied to an immunohistofluorescence setting.

In this manuscript, the authors focus on the use of quantum dot-conjugated antibodies in an immunohistology context. Here, samples are washed several times with ethanol and then infused with paraffin. The samples are then incubated with the conjugated antibodies and imaged. Was this technique was selected due to its ability to preserve the physical morphology of the food matrix? Is this useful from a food science point of view (ie: detecting the distribution and colocalization of allergenic proteins)? However, the suitability for allergen detection and quantification is not as clear, and could use some discussion. In particular, the repeated ethanol washes have the potential to remove allergen proteins, while the addition of paraffin to the matrix could inhibit antibody binding to whatever allergens remain. Indeed, the presence of the food matrix itself may not conducive to quantitative analysis due to its heterogeneous nature. In particular, variations in exposed surface area, particle size, and distribution represent confounding variables could make quantification difficult. Are there literature comparisons with homogenization used in a more conventional ELISA setup?  Could the authors calibrate their measurements with previous standards?

Additionally, there are concerns about the specific data presented in this paper, a further explanation of which is provided below.

More specific comments:

Materials and Methods:

The IHF-QD method section does not provide any detail as to how the samples were prepared for Simulation 2 – were they fixed and paraffin-infused as in Simulation 1? This is an important detail for reproducibility.

Results and Discussion:

Table 1: Table 1 provides a classification scheme of yes/no/duplicite(see below). However, no details are provided either here or on the methods section as to how these classifications were determined. In the methods section (lines 161-168) the authors indicate that the IHF-QD data was quantified by looking at both the mean signal intensity, as well as the generation of histograms for pixel intensity. Were either of these measures used to produce the classifications presented in Table 1? If so, what was the methodology and criteria for doing so? Was there an absolute cut-off value for signal intensity? If so, how was this cut-off value determined? This is confusing because line 214 states that the IHF-QD data is qualitative.

In Table 1 and elsewhere what does duplicite mean?  This is not a word in English.  Does it mean there were duplicate measurements?  Or the check/plus format seems to suggest that duplicate measurements gave different answers.  Please carefully explain the meaning of the word and symbology.

Figures 3,4:  The application of PCA is not obviously applicable here.  Although it is not totally clear from the description, the authors appear to be using replicate measurements from different techniques as the axis of separation.  PCA is applicable when multiple measurements are made on the same sample in order to understand which measurement(s) contribute most to differentiating the input.  It is not usually a statistical analysis of error or contrast.  The t-test is more appropriate here.

Figures 5, 6: These figures provide an effective visual illustrating the use of IHF-QD to identify and detect allergens. However, the authors should provide an image for a negative control (ie: IHF-QD of an AraK- and G K – sample). They should also provide a comparison image showing allergen detection via the ELISA method to illustrate the advantages of IHF-QD.

Table 2: Here the authors apply their IHF-QD method against commercially available food items, and compare the results against those obtained from ELISA. Here, the ELISA data is presented in a quantitative manner, with clear mg/kg cut-off values for the various classifiers (lines 327-329). However, again the materials and methods section does not provide any information as to how these quantitative results were obtained. Was an aqueous extract prepared from the commercial samples and analysed using immobilized antibodies as per a more traditional ELISA setup? What is the basis for comparing these results to those obtained from IHF-QD, which involve the imaging of fixed, solid, non-aqueous samples? How were the resulting values quantified (mean pixel intensity of the resulting images?), and how can they be accurately compared to the standard curve obtained from homogeneous, aqueous samples used to generate a standard curve for ELISA as described in lines 188-192? Additionally, as with Table 1, the authors report the IHF-QD data in the form of a dichotomous classification (+/-), but yet provide no details as to how this classification was determined. In other words, more details are needed to explain how the qualitative IHF-QD data was classified in an impartial manner.

Overall, the authors should include more data from their IHF-QD studies to support the results presented in tables 1 and 2, whether it is in the form of histograms, overall fluorescence intensity, etc. They also need to include data or images from their negative controls (IHF-QD on AraK- and G K -), as well as their ELISA’s to allow the reader to make a direct comparison between the two methods. More information is required as to the methods through which the ELISA data was obtained to give the reader confidence that the two methods are being compared in a fair and impartial manner. Additionally, more details are required as to the methods used to generate the ELISA data.

As mentioned in the overall comments section, there appears to be an opportunity to quantitatively compare the techniques, or calibrate the quantum dot measurements against traditional ELISA techniques. However, additional data and possibly experiments demonstrating a clear and tangible comparison of IHF-QD, or even QD-conjugated antibodies to more traditional plate-ELISA application to detect and quantify allergens in a range of consumer food products would address these concerns.

Secondary issues:

Table 1 and elsewhere: The authors refer to samples AraK1+, Ara K2+…etc. and G K1, G K2…etc. throughout their manuscript. However, it is never indicated exactly what concentration of allergens these samples correspond to

Figures 1 and 2: This figure shows the response of the various QD-conjugated antibodies to gluten and peanut allergens. However, it is not indicated which samples (eg: AraK1, G K 2…etc.) was used to generate these values. Also, there is no Y-axis label.

Lines 280-283: Not clear how these advantages are relevant to the detection of allergens.

Minor issues:

 Numerous grammatical errors. The use of an English editing service is recommended. Examples of such errors include, but are not limited to:

Line 17: the use of the word “simulations” provides the wrong connotations – suggests in silicoexperiments/simulations. Line 26 “slightly differences” Line 139 “with samples were microwave heating 650 W for 5 min, then cooling….” Line 160 “Comparison leads to find differences in results…” Line 168 “bylo provedeno” is Czech for “Was carried out” Line 184 “tested during test development”. Not clear what this means. Was this tested by the manufacturer, or by the authors? Line 269 “important answers to our results”

Line 72: “Chromogenes” is misspelt. Should be “Chromogens”.

Line 74-75: Are fluorophores quenched under normal imaging conditions? The study cited describes a competitive ELISA assay and does not mention quenching. Quenching occurs when there is a high local concentration of fluorophores of any type – QD’s are not immune from this either.

Line 85: It is not clear what the authors mean by “Absorption”. Perhaps “Quantum Yield” is a more accurate term

Lines 102-103: As discussed before, the authors should define AraK1-AraK8. What are the concentrations of peanut matter for each of these samples.

Line 130: The authors state that the samples had to be “further treated because they were subject to various…” What does this “further treatment” entail?

Line 131: It is not clear what is meant by “this could overlap antigens with other molecules”. Do the authors refer to the ability of the food matrix to prevent antibody binding. If so, this raises further questions as to the suitability of IFH methods to detect and quantify allergens in commercial food samples

Line 330: The authors indicate a false negative rate of 25% for manufacturer self-identification. How does this compare to rates of false negatives reported in other papers? It should be noted that Pele et al. (010) also report a similar false negative rate (~25%) using ELISA.

Author Response

Reviewer 1

In this manuscript, Ludmila et al. report on the use of quantum dots in immunohistofluorescence (IHF-QD) to detect allergens in food samples. Food allergen detection is a major issue in public health and new approaches are welcome. While the authors appear successful in this endeavor, they fail to provide quantitative comparisons of their IHF-QD technique over existing methodologies. Quantum dots have been used previously to detect food allergens (including some of the same allergens presented in this study), and without the quantitative comparison, the utility of the paper is questionable as it appears to be repetitive.

Our article is not repetitive compared to other previous studies - e.g. the study Ansari et al.[44] - it deals with similar issue, but only the distribution of gliadin in model dough and model baked bread samples. Gliadin distribution was monitored by CLSM. In contrast, our study is devoted to the detection of allergens (wheat / peanut), from model samples and from samples from the market also. Our study aims to show that the IHF-QD method is applicable not only to the analysis of model samples of known composition, but also to the analysis of food samples which are subject to different technological processing in production and have different composition. Furthermore, the equipment we use for imaging - a fluorescent microscope - is simpler and less expensive.

While the authors accurately identify several advantages of quantum dots over traditional fluorophores (increased quantum yield, stability etc.), they fail to provide any quantitative comparison to demonstrate these advantages when applied to an immunohistofluorescence setting.

Overall achievement of our study is the finding that IFH-QD method gives at least the same results as ELISA method. The study is giving the possibility for applying another kind of method for food allergen detection. Certainly, that future studies will show more exact how this kind of method should be applied. Consequently, our research represents a valuable source for the application of IFH-QD methods.

In this manuscript, the authors focus on the use of quantum dot-conjugated antibodies in an immunohistology context. Here, samples are washed several times with ethanol and then infused with paraffin. The samples are then incubated with the conjugated antibodies and imaged. Was this technique selected due to its ability to preserve the physical morphology of the food matrix?

The methodology was chosen in order to preserve the sample matrix structures. Without a compact matrix, immunohistochemical / immunohistofluorescence methods are almost impossible to perform - the integrity of the histological cut is important. Moreover, based on the morphological structure, we are able to really classify that it is a sought / analyzed component.

Our research was mainly focused on allergens detection, though we observed different quantum yield intensity among samples, same as between different Qdots conjugates. These findings, together with propriate software for the evaluation, will be ceirtainly beneficial for future studies of allergen quantification.

In this manuscript, the authors focus on the use of quantum dot-conjugated antibodies in an immunohistology context. Here, samples are washed several times with ethanol and then infused with paraffin. The samples are then incubated with the conjugated antibodies and imaged. Was this technique was selected due to its ability to preserve the physical morphology of the food matrix? Is this useful from a food science point of view (ie: detecting the distribution and colocalization of allergenic proteins)? However, the suitability for allergen detection and quantification is not as clear, and could use some discussion. In particular, the repeated ethanol washes have the potential to remove allergen proteins, while the addition of paraffin to the matrix could inhibit antibody binding to whatever allergens remain.

Histological processing of samples for IHC and IHF methods involves ethanol washing, as you type, which may potentially cause allergen removal from the sample matrix, but we have accredited methods at our UVPS in an accredited laboratory, the procedure also involving washing the sample with ethanol. Therefore, we use the antigen-retrieval step within these methods (and ours too) - we apply e.g. citrate buffer (in our study citrate EDTA or AlCl3). Our methodology also includes permeabilization in 0.25% Triton X-100, which Triton X-100 is commonly used for the isolation of membrane protein complexes, and also for most similar surfactants. Triton X-100 is often used for co-immunoprecipitation experiments also.

As for paraffin and problems, inhibition of antibody binding to remaining allergens – to prevent nonspecific binding in our method we used blocking buffer 6% BSA with goat normal serum. We describe it in IHF-QD microscopic method part and also we emphasiyed in Preparation and processing of samples part.

We are also attaching  You a link with the certification of our immunohistochemical laboratory:

Indeed, the presence of the food matrix itself may not conducive to quantitative analysis due to its heterogeneous nature. In particular, variations in exposed surface area, particle size, and distribution represent confounding variables could make quantification difficult. Are there literature comparisons with homogenization used in a more conventional ELISA setup? Could the authors calibrate their measurements with previous standards?

We agree with reviewer, but ELISA analysis kits were provided by the manufacturer and we conducted analyses according to manufacturer’s manual and instructions.

More specific comments:

Materials and Methods:

The IHF-QD method section does not provide any detail as to how the samples were prepared for Simulation 2 – were they fixed and paraffin-infused as in Simulation 1? This is an important detail for reproducibility.

Samples prepared for simulation 2 were fixed and soaked with paraffin as for Simulation 1. The explenation of sample preparation for Simulation 2 is added to the manuscript.

Results and Discussion:

Table 1: Table 1 provides a classification scheme of yes/no/duplicite(see below). However, no details are provided either here or on the methods section as to how these classifications were determined.

The word duplicite - maybe not properly used, for which we apologize. Rather, the word doubtful should be used. We discribe it now more preciously in the manuscript – IHF-QD microscopic method.

In the methods section (lines 161-168) the authors indicate that the IHF-QD data was quantified by looking at both the mean signal intensity, as well as the generation of histograms for pixel intensity. Were either of these measures used to produce the classifications presented in Table 1? If so, what was the methodology and criteria for doing so? Was there an absolute cut-off value for signal intensity? If so, how was this cut-off value determined? This is confusing because line 214 states that the IHF-QD data is qualitative.

The signal intensity (quantum yield) evaluation is not related to the evaluation of the results in Table 1. The signal intensity evaluation is only for selecting / evaluating the appropriate commercial conjugate.

In Table 1 and elsewhere what does duplicite mean? This is not a word in English.  Does it mean there were duplicate measurements?  Or the check/plus format seems to suggest that duplicate measurements gave different answers.  Please carefully explain the meaning of the word and symbology.

It is already explained.

Figures 3,4: The application of PCA is not obviously applicable here.  Although it is not totally clear from the description, the authors appear to be using replicate measurements from different techniques as the axis of separation.  PCA is applicable when multiple measurements are made on the same sample in order to understand which measurement(s) contribute most to differentiating the input.  It is not usually a statistical analysis of error or contrast.  The t-test is more appropriate here.

We included ONEWAY ANOVA test.

Figures 5, 6: These figures provide an effective visual illustrating the use of IHF-QD to identify and detect allergens. However, the authors should provide an image for a negative control (ie: IHF-QD of an AraK- and G K – sample). They should also provide a comparison image showing allergen detection via the ELISA method to illustrate the advantages of IHF-QD.

It is not possible; or rather appropriate, to provide a negative control figure, since the reader would not see anything but a “black” figure. We do not have a comparative figure showing the detection of allergens by ELISA, we measured on a reader and did not take a photo of a plate to avoid breaking the manufacturer's methodology.

Table 2: Here the authors apply their IHF-QD method against commercially available food items, and compare the results against those obtained from ELISA. Here, the ELISA data is presented in a quantitative manner, with clear mg/kg cut-off values for the various classifiers (lines 327-329). However, again the materials and methods section does not provide any information as to how these quantitative results were obtained. Was an aqueous extract prepared from the commercial samples and analyzed using immobilized antibodies as per a more traditional ELISA setup? What is the basis for comparing these results to those obtained from IHF-QD, which involve the imaging of fixed, solid, non-aqueous samples? How were the resulting values quantified (mean pixel intensity of the resulting images?), and how can they be accurately compared to the standard curve obtained from homogeneous, aqueous samples used to generate a standard curve for ELISA as described in lines 188-192? Additionally, as with Table 1, the authors report the IHF-QD data in the form of a dichotomous classification (+/-), but yet provide no details as to how this classification was determined. In other words, more details are needed to explain how the qualitative IHF-QD data was classified in an impartial manner.

The quantitative results of the ELISA methods were obtained by measuring the ELISA reader according to the instructions of the ELISA kit manufacturer and the associated program. This associated program can automatically evaluate the content of a given allergen in mg/kg based on standards measurement. These values can also be verified by calculations according to the obtained calibration curve. The better explanation is now provided with in the manuscript.

The pixel measurement in case of IHF-QD method is not related to the evaluation of the contained allergen in mg/kg. The quantitative evaluation in mg/kg relates only to the ELISA method.

The qualitative results of the IHF-QD method were investigated anonymously, it means – the samples for IHF-QD analysis had their own different coding from the samples for ELISA method. The results were matched finally. For the sake of clarity, a united simpler coding was used for this article.

Overall, the authors should include more data from their IHF-QD studies to support the results presented in tables 1 and 2, whether it is in the form of histograms, overall fluorescence intensity, etc. They also need to include data or images from their negative controls (IHF-QD on AraK- and G K -), as well as their ELISA’s to allow the reader to make a direct comparison between the two methods. More information is required as to the methods through which the ELISA data was obtained to give the reader confidence that the two methods are being compared in a fair and impartial manner. Additionally, more details are required as to the methods used to generate the ELISA data.

As mentioned in the overall comments section, there appears to be an opportunity to quantitatively compare the techniques, or calibrate the quantum dot measurements against traditional ELISA techniques. However, additional data and possibly experiments demonstrating a clear and tangible comparison of IHF-QD, or even QD-conjugated antibodies to more traditional plate-ELISA application to detect and quantify allergens in a range of consumer food products would address these concerns.

As mentioned above, negative control figures can’t be included because nothing would be evident from it, just black figure. We agree with the reviewer that our study should lead to allergen quantification, but this is our first part of research considering these issues. In our ongoing research and planned in the future we will certainly be more able to quantify our results.

Secondary issues:

Table 1 and elsewhere: The authors refer to samples AraK1+, Ara K2+…etc. and G K1, G K2…etc. throughout their manuscript. However, it is never indicated exactly what concentration of allergens these samples correspond to.

Ara K1 to AraK8 or GK1 to GK8 are model samples, always repeated twice, the allergenic component has been added in a growing concentration series 0,01 %, 0,1 %,  1%,10 %. We included the table (new Table 1) with samples explanation.

Figures 1 and 2: This figure shows the response of the various QD-conjugated antibodies to gluten and peanut allergens. However, it is not indicated which samples (eg: AraK1, GK 2…etc.) was used to generate these values. Also, there is no Y-axis label.

These are model samples with the 1 % addition of allergenic components. We repare the name of Figure 5 and Figure 6. Now it is clearer, from which sample the picture was captured.

Lines 280-283: Not clear how these advantages are relevant to the detection of allergens.

We agree with the reviewer. Thank You for notice, we remove misleading informations from manuscript.

Minor issues:

Numerous grammatical errors. The use of an English editing service is recommended. Examples of such errors include, but are not limited to: Line 17: the use of the word “simulations” provides the wrong connotations – suggests in silicoexperiments/simulations. Line 26 “slightly differences” Line 139 “with samples were microwave heating 650 W for 5 min, then cooling….” Line 160 “Comparison leads to find differences in results…” Line 168 “bylo provedeno” is Czech for “Was carried out” Line 184 “tested during test development”. Not clear what this means. Was this tested by the manufacturer, or by the authors? Line 269 “important answers to our results”.

It was repaired. We rewrote the majority of the manuscript and we hope that now it written in more appropriate way.

Line 72: “Chromogenes” is misspelt. Should be “Chromogens”.

It was repaired.

Line 74-75: Are fluorophores quenched under normal imaging conditions? The study cited describes a competitive ELISA assay and does not mention quenching. Quenching occurs when there is a high local concentration of fluorophores of any type – QD’s are not immune from this either.

It was repaired.

Line 85: It is not clear what the authors mean by “Absorption”. Perhaps “Quantum Yield” is a more accurate term

It was repaired.

Lines 102-103: As discussed before, the authors should define AraK1-AraK8. What are the concentrations of peanut matter for each of these samples?

It was repaired.

Line 130: The authors state that the samples had to be “further treated because they were subject to various…” What does this “further treatment” entail?

This sentence was erased from the manuscript; we agree with reviewer that this sentence is misleading.

Line 131: It is not clear what is meant by “this could overlap antigens with other molecules”. Do the authors refer to the ability of the food matrix to prevent antibody binding. If so, this raises further questions as to the suitability of IFH methods to detect and quantify allergens in commercial food samples

This sentence was erased from the manuscript; we agree with reviewer that this sentence is misleading.

Line 330: The authors indicate a false negative rate of 25% for manufacturer self-identification. How does this compare to rates of false negatives reported in other papers? It should be noted that Pele et al. (010) also report a similar false negative rate (~25%) using ELISA.

It was repaired in manuscript.

Reviewer 2 Report

The manuscript “Use of IHF-QD microscopic analysis for the detection of food allergenic components: peanuts and gluten” regards an important topic in the field of food allergy, namely the development of methods for the detection of allergenic ingredients in complex processed foods. The manuscript proposes an interesting approach to this theme. Although the proposed method is not quantitative, it seems a valuable qualitative strategy. In general, the manuscript is well written, it presents the results in an adequate form, which are also well discussed with the available literature. Literature revision also seems well conducted.

Food allergens regards to the allergenic proteins (Ara h 1 in peanut and gliadin in wheat), while peanut and wheat are allergenic foods/ingredients. The authors often confuse/mistake these concepts. Please revise the manuscript accordingly to clarify the text. In fact, authors target the gliadin, which is a fraction of gluten, but it is also wheat. Please consider changing the title and all additional sections to wheat instead of gluten. Wheat is an allergenic ingredient/food, but gluten is a protein fraction. Regulation and food manufacturers must inform on the allergenic ingredients/foods potentially allergenic, not on food allergens (as proteins). This should also be revised along the manuscript (example: line 41). Lines 48-52 - the sentence is too confusing. Please rewrite to clarify its meaning. When describing the current immunoassays, PCR and MS methods for the detection of food allergenic foods, use more recent publications or even review papers. Or maybe focused it on peanut and wheat. Section “The analyzed material” How do the authors prepare the model mixtures? How do they ensure their homogeneity during model mixture preparation and then during IHF-QD microscopic analysis? 0.001% is 10 mg of allergenic ingredient/kg of matrix, so how was this model sample prepared to ensure complete homogeneity of the pastry? Where the samples grid before analysis? Section “Preparation and processing of samples” Xylene and alcohol baths were used. Do the authors know if conformational alterations can be induced by these reagents to the allergenic proteins present in foods? Line 137-138 – please provide more information on the buffers (concentration, pH). Figure 1 and 2 – explain such huge standard deviation (D and C, respectively), because it suggests lack of precision and also some accuracy to the method. Figure 1 and 2 – statistical analysis is not presented in both figures. Please add. Line 258-260 – “The results show that the QD-IHF microscopic method using QD with 585 nm emission achieves the highest contrast intensity compared to the same method using QDs with emission 525 nm or 655 nm.”, but again lacks explanation on precision and accuracy (high standard deviation). Table 2 – present quantification data for ELISA. If negative indicate

Author Response

Reviewer 2

The manuscript “Use of IHF-QD microscopic analysis for the detection of food allergenic components: peanuts and gluten” regards an important topic in the field of food allergy, namely the development of methods for the detection of allergenic ingredients in complex processed foods. The manuscript proposes an interesting approach to this theme. Although the proposed method is not quantitative, it seems a valuable qualitative strategy. In general, the manuscript is well written, it presents the results in an adequate form, which are also well discussed with the available literature. Literature revision also seems well conducted.

Food allergens regards to the allergenic proteins (Ara h 1 in peanut and gliadin in wheat), while peanut and wheat are allergenic foods/ingredients. The authors often confuse/mistake these concepts. Please revise the manuscript accordingly to clarify the text. In fact, authors target the gliadin, which is a fraction of gluten, but it is also wheat. Please consider changing the title and all additional sections to wheat instead of gluten. Wheat is an allergenic ingredient/food, but gluten is a protein fraction.

We used vital wheat protein and peanut seeds in our study. The title was adjusted and the manuscript was repaired too.

Regulation and food manufacturers must inform on the allergenic ingredients/foods potentially allergenic, not on food allergens (as proteins). This should also be revised along the manuscript (example: line 41). Lines 48-52 - the sentence is too confusing.

It was repaired.

Please rewrite to clarify its meaning. When describing the current immunoassays, PCR and MS methods for the detection of food allergenic foods, use more recent publications or even review papers. Or maybe focused it on peanut and wheat.

We added additional references and we hope that our results are now more connected with previously published works.

Section “The analyzed material” How do the authors prepare the model mixtures? How do they ensure their homogeneity during model mixture preparation and then during IHF-QD microscopic analysis? 0.001% is 10 mg of allergenic ingredient/kg of matrix, so how was this model sample prepared to ensure complete homogeneity of the pastry? Where the samples grid before analysis?

Model samples were prepared with addition of wheat protein or grinded peanut seeds in increased concentration from 0.01 % to 10 %. The wheat protein or grinded peanut seed in desired concentration was added to dough (consist from sugar, corn flour, egg, milk and baking powder). Dough was stirring by dough kneader. Dough with one rate of concentration was split into smaller parts of fist size and baked at 180°C for 15 minutes in oven.

Section “Preparation and processing of samples” Xylene and alcohol baths were used. Do the authors know if conformational alterations can be induced by these reagents to the allergenic proteins present in foods?

Thank you for your inquiry. Yes, we know, therefore, our analysis involves steps like the antigen-retrieval in citrate EDTA or AlCl3 buffer. Our methodology also includes permeabilization in 0.25% Triton X-100 and step for prevention of nonspecific binding used blocking buffer 6% BSA with goat normal serum. We describe it in IHF-QD microscopic method part and also we emphasied in Preparation and processing of samples part.

We are also attaching you a link with the certification of our immunohistochemical laboratory:

Line 137-138 – please provide more information on the buffers (concentration, pH).

It is described in manuscript now. More information about citrate EDTA: http://www.ihcworld.com/_protocols/epitope_retrieval/citrate_edta.htm

Figure 1 and 2 – explain such huge standard deviation (D and C, respectively), because it suggests lack of precision and also some accuracy to the method.

We agree with the reviewer that the deviations are large and therefore the method should be more standardized. Our results show the complexity of allergen detection in food. P 585 has the highest deviation, but shows the highest radiation intensity in microscopic analysis.

Figure 1 and 2 – statistical analysis is not presented in both figures. Please add. Line 258-260 – “The results show that the QD-IHF microscopic method using QD with 585 nm emission achieves the highest contrast intensity compared to the same method using QDs with emission 525 nm or 655 nm.”, but again lacks explanation on precision and accuracy (high standard deviation).

It was explained in manuscript.

Table 2 – present quantification data for ELISA. If negative indicate It was repaired.

Line 337-339 – change accordingly “Despite a few promising treatment strategies in the area of food allergies, treatment is also largely based on the eviction of allergen-containing foods.

Thank You, It was repaired.

Lines 357-359 – sentence to confusing, please rewrite to clarify its meaning.

It was repaired.

Lines 359-365 – if possible, provide world’s overview on hospital admission due to food allergies, not only the USA.

We added additional references.

Please give an estimative of the time/cost (including equipment/reagents) per analysis using the proposed IHF-QD microscopic method compared to the current immunoassays, PCR or even MS methods. Please realize that at this point the price per allergen analysis by PCR and ELISA are quite similar and the less expensive compared to MS methods. How would you include the proposed IHF-QD microscopic method compared to the remaining methods? Is it possible to implement the propose IHF-QD microscopic method at a routine analysis basis? Line 401 – “…IHF-QD microscopic method can be considered a reliable tool for the rapid detection of gliadin and…” How rapid?

Certainly, at this point the proposed IHF_QD microscopic method cannot be proposed as routine food allergen analysis. The cost of the method on which we based our research is approximately the same as ELISA. On the other hand, our research is showing the possibility of using IHF-QD method as one option for food allergen detection. The advantage of IHF-QD method is possibility to control the obtained result based on knowledge of food component microstructure.  New techniques for food allergen detection are necessary because they can represent little step toward achievement of more reliable and at the end less time consuming and less expensive method.

Reviewer 3 Report

The purpose of the study is poorly presented. It is not clear what advantage was obtained by conducting the histological analyses inasmuch as the results did not exceed those generated using the ELISAs. Indeed, the ELISAs provided also quantitative results which are critical in the detection of food allergens and gluten; the later having a regulatory target level of 20 ppm in the EU, USA, and Canada.

It is strongly recommended that the manuscript should be re-written,

given a better descriptive title (that includes 'histological'),

with a refocus to emphasize what is important / goal of using such an assay.

If the multiplex ability associated with the different emission spectra is the point, then illustrate such. In doing such, real-life / prepared food samples must be used as examples prepared from the actual allergenic foods and not purified proteins inasmuch as the laws of the EU and USA deal with the allergenic foods and their concentrations.

However, as currently written, it is not clear what the purpose of the manuscript is, especially since other papers have been published using QDs for histolographic analyses.

Author Response

Reviewer 3

The purpose of the study is poorly presented. It is not clear what advantage was obtained by conducting the histological analyses inasmuch as the results did not exceed those generated using the ELISAs. Indeed, the ELISAs provided also quantitative results which are critical in the detection of food allergens and gluten; the later having a regulatory target level of 20 ppm in the EU, USA, and Canada.

It is strongly recommended that the manuscript should be re-written, given a better descriptive title (that includes 'histological'), with a refocus to emphasize what is important / goal of using such an assay.

Thank You for Your suggestions we made major revisions of our manuscript and we hope that results obtained by our research are better represented.

If the multiplex ability associated with the different emission spectra is the point, then illustrate such. In doing such, real-life / prepared food samples must be used as examples prepared from the actual allergenic foods and not purified proteins inasmuch as the laws of the EU and USA deal with the allergenic foods and their concentrations.

We changed the term in the manuscript same as in the title by changing gluten to wheat protein. With this change the nature of our experiment is clearer. We would also like to add that vital wheat protein is often used in bakery production as fortifying element.

However, as currently written, it is not clear what the purpose of the manuscript is, especially since other papers have been published using QDs for histolographic analyses.

The purpose of the study was to check the possibility of food allergen determination by IHF-QD method. Though, this method has been apply before, we still think that more experiments are necessary for finding reliable food allergen detection method. Developed method will be less costly, time consuming, and at the same time high sensitive. Considering these facts, our study will certainly help future works to at least modulate experiments in slightly different way from our, and maybe be closer to the mentioned discovery.

Round 2

Reviewer 1 Report

The authors have better defined their qualitative method over the more traditional quantitative ELISA.  The authors should be encouraged in follow up studies to develop a more quantitative methodology.

Several items must be fixed prior to acceptance: 

Apparently, it was not explicitly clear in the previous remarks that the PCA method in figures 3 and 4 is an incorrect application of PCA and must be removed.  The t-test or ANOVA as added by the authors is better.  However the results of the test are not described.

The authors refuse to show to a negative control experiment in figures 5 and 6 because they assert that data would be a blank or black image.  That is precisely the point of a negative control experiment.  This could be presented adjacent to a light microscope image of the same sections to show what the picture was taken of.

The authors misunderstood the intent of the comments regarding the ELISA.  There was no desire to show an image of the ELISA plate, just an interest in showing the ELISA data (concentration versus intensity). This would be to demonstrate that 1. The data was acquired. 2. The authors got clean data in a measurable range. Etc.  Supplemental figures would be adequate if the authors wish to highlight their QD technology in main document. The comments that all the ELISA experiments were run according to manufacturer instructions is not adequate.

Author Response

The authors have better defined their qualitative method over the more traditional quantitative ELISA.  The authors should be encouraged in follow up studies to develop a more quantitative methodology.

Several items must be fixed prior to acceptance: 

1) Apparently, it was not explicitly clear in the previous remarks that the PCA method in figures 3 and 4 is an incorrect application of PCA and must be removed.  The t-test or ANOVA as added by the authors is better.  However the results of the test are not described.

  • PCA figures are deleted, and we also included more comments about results gained ANOVA.

2) The authors refuse to show to a negative control experiment in figures 5 and 6 because they assert that data would be a blank or black image.  That is precisely the point of a negative control experiment.  This could be presented adjacent to a light microscope image of the same sections to show what the picture was taken of.

  • The figure of negative control obtained by IHF-QD method was added.

3) The authors misunderstood the intent of the comments regarding the ELISA.  There was no desire to show an image of the ELISA plate, just an interest in showing the ELISA data (concentration versus intensity). This would be to demonstrate that 1. The data was acquired. 2. The authors got clean data in a measurable range. Etc.  Supplemental figures would be adequate if the authors wish to highlight their QD technology in main document. The comments that all the ELISA experiments were run according to manufacturer instructions is not adequate.

  • Qualitative, not quantitative indicators were compared. For the conversion we use our own algorithms where the sample is not marked and is anonymously evaluated in the program so that the results are as accurate as possible. The ELISA plate is evaluated by a spectrophotometric instrument (ELISA reader-Sunrise TECAN), photo documentation is not a common part of the ELISA plate evaluation procedure. Yes, we agree with reviewer, the picture would be very illustrative and would certainly be an integral part of the presentation of the results at conferences. In case there is a great reader's interest in illustrative description and methodical procedure, there is no problem to establish contact within the framework of cooperation between institutions.
  • Quantitative data obtained by ELISA were added to the Table 3.

Reviewer 2 Report

The authors have improved some parts of the manuscript but there were still some unanswered questions. Please provide clear information on the following topics.

Section “The analyzed material” How do the authors prepare the model mixtures? What was the final amount of dough (1 kg, 100 g or 10g ???) prepare for each concentration? Were the model samples prepared individually? For the pastry with 0.01% of peanut, what were the quantities used? Example: 100 mg peanut to 1 kg of pastry? How can the authors prepare model samples with increasing percentages? By performing it individually? This must be clear in the respective section, since it is mandatory information for other authors being capable of reproducing the results!!! “When describing the current immunoassays, PCR and MS methods for the detection of food allergenic foods, use more recent publications or even review papers. Or maybe focused it on peanut and wheat.”The authors did not address this comment. In the revised version (v2), the papers cited are exactly the same as version 1. What was changed? It is not possible to identify literature review based on the changes. Why citing a paper on sesame if there are examples for peanut and wheat in the literature? “Table 2 – present quantification data for ELISA. If negative indicate

Author Response

The authors have improved some parts of the manuscript but there were still some unanswered questions. Please provide clear information on the following topics.

1) Section “The analyzed material” How do the authors prepare the model mixtures? What was the final amount of dough (1 kg, 100 g or 10g ???) prepare for each concentration? Were the model samples prepared individually? For the pastry with 0.01% of peanut, what were the quantities used? Example: 100 mg peanut to 1 kg of pastry? How can the authors prepare model samples with increasing percentages? By performing it individually? This must be clear in the respective section, since it is mandatory information for other authors being capable of reproducing the results!!!

  • Than You for Your feedback, we agree with it. We added a new part to section Material and methods.

2) “When describing the current immunoassays, PCR and MS methods for the detection of food allergenic foods, use more recent publications or even review papers. Or maybe focused it on peanut and wheat.”The authors did not address this comment. In the revised version (v2), the papers cited are exactly the same as version 1. What was changed? It is not possible to identify literature review based on the changes. Why citing a paper on sesame if there are examples for peanut and wheat in the literature?

  • We added new references to themanuscript.

3) “Table 2 – present quantification data for ELISA. If negative indicate

  • Following LOD concentrations were added under Table 2:

LOD peanut: 0.03 – 0.13 mg/kg

LOD gliadin: 1.2 – 1.5 mg/kg

  • Quantitative data completed in Table 3, values above the LOQ limit marked greater than 80 (gluten) or 25 (peanut).

4) Lines 373-375– the sentence is still not correct. “The American population suffers from 3.5 % to 4 % of the total population, with 4 % for adults and 5% to 8% for children.” Suffers from what? Did the authors read the revised sentence???

  • The sentence was repaired, we apollogize for this mistake.

5) Reference 55 is not correctly presented. Please revise.

  • The reference was repaired.

6) References 57 and 58 do not seem adequate to support the sentence in lines 386-387. Please revise. Literature suggestions: https://doi.org/10.1111/j.1398-9995.2010.02346.x., https://doi.org/10.1111/all.12341. Or similar.

  • We added the proposed literature.

7) “Please give an estimative of the time/cost (including equipment/reagents) per analysis using the proposed IHF-QD microscopic method compared to the current immunoassays, PCR or even MS methods. Please realize that at this point the price per allergen analysis by PCR and ELISA are quite similar and the less expensive compared to MS methods. How would you include the proposed IHF-QD microscopic method compared to the remaining methods? Is it possible to implement the propose IHF-QD microscopic method at a routine analysis basis?” The authors do not fully answer to this comment. Please add information on cost per analysis, time per analysis and compare with other methods. This must be added in the conclusion section of the manuscript, evidencing how the proposed IHF-QD microscopic method can be considered as a potential alternative to the other techniques. “Line 419 – “…IHF-QD microscopic method can be considered a reliable tool for the rapid detection of gliadin and…” How rapid?” if the authors address the comment above it will also answer to this question.

  • We added our statements in the conclusion part.

8) Acronyms should not be used in plural. Example QDs??

  • Thank You, we repaired it in manuscript.

Reviewer 3 Report

re-read to improve phraseology

Author Response

1) Re-read to improve phraseology.

- Thank You very much for the very helpful comments and suggestions. We tried to modify the phraseology and style of English.